Who’s your mama? Riverine hybridisation of threatened freshwater Trout Cod and Murray Cod

Couch Alan J. alan.couch@canberra.edu.au
Unmack Peter J.
Dyer Fiona J.
Lintermans Mark
Institute for Applied Ecology, University of Canberra , Bruce , ACT , Australia
Garant Dany
Electronic publication date: 2016 Oct 27
Publication date: 2016
Volume: 4
Electronic Location ID: e2593
Received 2016 Jun 14; Accepted 2016 Sep 20
Copyright: ©2016 Couch et al.
Copyright year: 2016
Copyright holder: Couch et al.
License: This is an open access article distributed under the terms of the Creative Commons Attribution License, which permits unrestricted use, distribution, reproduction and adaptation in any medium and for any purpose provided that it is properly attributed. For attribution, the original author(s), title, publication source (PeerJ) and either DOI or URL of the article must be cited.
License URL: https://creativecommons.org/licenses/by/4.0/

Keywords: Maccullochella, Hybridisation, Murrumbidgee River, Trout Cod, Murray Cod, Introgression, Restocking, SNPs, Conservation genetics, Freshwater management

Funding: Australian Government Department of Education and Training ICON Water Recreational Fishing Research The Australian Government Department of Education and Training and ICON Water (ACT) provided funding for this project as a PhD student scholarship. Additional financial support was provided by Carlton-Mid through Recreational Fishing Research (http://recfishingresearch.org). The funders had no role in study design, data collection and analysis, decision to publish, or preparation of the manuscript.

==============================
Rates of hybridization and introgression are increasing dramatically worldwide because of translocations, restocking of organisms and habitat modifications; thus, determining whether hybridization is occuring after reintroducing extirpated congeneric species is commensurately important for conservation. Restocking programs are sometimes criticized because of the genetic consequences of hatchery-bred fish breeding with wild populations. These concerns are important to conservation restocking programs, including those from the Australian freshwater fish family, Percichthyidae. Two of the better known Australian Percichthyidae are the Murray Cod, Maccullochella peelii and Trout Cod, Maccullochella macquariensis which were formerly widespread over the Murray Darling Basin. In much of the Murrumbidgee River, Trout Cod and Murray Cod were sympatric until the late 1970s when Trout Cod were extirpated. Here we use genetic single nucleotide polymorphism (SNP) data together with mitochondrial sequences to examine hybridization and introgression between Murray Cod and Trout Cod in the upper Murrumbidgee River and consider implications for restocking programs. We have confirmed restocked riverine Trout Cod reproducing, but only as inter-specific matings, in the wild. We detected hybrid Trout Cod–Murray Cod in the Upper Murrumbidgee, recording the first hybrid larvae in the wild. Although hybrid larvae, juveniles and adults have been recorded in hatcheries and impoundments, and hybrid adults have been recorded in rivers previously, this is the first time fertile F1 have been recorded in a wild riverine population. The F1 backcrosses with Murray cod have also been found to be fertile. All backcrosses noted were with pure Murray Cod. Such introgression has not been recorded previously in these two species, and the imbalance in hybridization direction may have important implications for restocking programs.

Introduction

Hybridization and introgression in the wild occurs in more than 10% of animal species, and is most common in more recently diverged species (Mallet, 2005). Rates of hybridization and introgression are increasing dramatically worldwide because of translocations, restocking of organisms and habitat modifications (Allendorf et al., 2001). Determining whether hybridization is occurring when reintroducing extirpated congeneric species is therefore important for recovery programs, where the goal is the conservation of the species through establishing additional self-sustaining populations, rather than improving fitness by introgression.

Introgression is now seen as an important phenomenon in many taxa, contributing to adaptation and speciation in plants, fish, and insects (Dowling & Secor, 1997; Baack & Rieseberg, 2007). While introgression can initially increase genetic diversity (Grossen et al., 2014) and adaptive introgression may be important in response to change (Hamilton & Miller, 2016), introgression can eventually reduce genetic diversity of one or both of the parent species, particularly if they are not naturally sympatric and one is introduced (Laikre et al., 2010). The genetic consequences of introgression are of increasing interest to conservation biologists as many species are on an irreversible path to extinction (the extinction vortex) (Frankham, Ballou & Briscoe, 2002), which can be initiated without any obvious signs (Blomqvist et al., 2010; Fagan et al., 2005). Furthermore, the synergistic effects of multiple extinction drivers, including genetic consequences, are only starting to be understood (Brook, Sodhi & Bradshaw, 2008). Introgression can be one of those drivers (Seehausen, 2006).

Restocking programs are questioned and often criticized because of potential genetic consequences of hatchery-bred fish breeding with wild populations, and such concerns are of critical importance to conservation restocking programs and in particular the potential changes to genetic diversity. For instance Allendorf et al. (2001) discusses introgressive hybridization as a particular genetic problem for native trout species in the USA. Laikre et al. (2010) provides examples of the problems from introductions of conspecifics in fishes, and Nock et al. (2011) describe adverse effects of stocking on the Australian Eastern Freshwater Cod (Maccullochella ikei), another endangered Australian percichthyid, including loss of heterozygosity and allelic richness. The particular case of potential introgressive hybridization arising from restocking an extirpated congeneric that have previously been partially sympatric is the focus of this study.

Two of the better known Australian Percichthyidae are Murray Cod (Maccullochella peelii) (Mitchell, 1938) and Trout Cod (Maccullochella macquariensis) (Cuvier, 1829). These iconic Australian species (only formally recognized as morphologically and genetically separate species in 1972 and 1978 respectively) (Berra & Weatherley, 1972; MacDonald, 1978) are two of four morphologically cryptic but genetically distinct species within Maccullochella (Rowland, 1993; Nock et al., 2010). Both species were highly sought after by anglers and previously overfished by commercial fishing until populations became so low the industry collapsed. The Murray Cod is Australia’s largest freshwater fish and can grow to as large as 180 cm in length. Both Trout Cod and Murray Cod are large-bodied species reaching maximum recorded weights of 16 and 113.6 kg, respectively, with fecundity ranging from 1,200-11,000 and 9,000-120,000 annual eggs in Trout Cod and Murray Cod respectively. Both are limited to parts of the Murray-Darling Basin (MDB) where the Trout Cod is endangered, and the Murray Cod is listed as vulnerable under the Australian Environment Protection and Biodiversity Conservation Act (1999) (Department of Environment, 2016). Murray Cod still form an important recreational fishery (Lintermans & Phillips, 2005; Ye et al., 2014; Koehn & Todd, 2012), whereas Trout Cod have been protected from recreational take since the early 1990s (Trout Cod Recovery Team, 2008).

Trout Cod were originally widely distributed in the Murray-Darling Basin (Lintermans, 2007), but by the 1980s had been reduced to just one single natural wild population and two translocated populations (Koehn et al., 2013). Trout Cod was one of the first Australian freshwater species identified and listed as threatened, with recovery actions now spanning more than 25 years (Koehn et al., 2013). In much of the Murrumbidgee River, Trout Cod and Murray Cod were sympatric until the late 1970s when Trout Cod were extirpated (Lintermans, Kukolic & Rutzou, 1988). Both Trout Cod and Murray Cod have active national recovery programs aimed at conserving both species as distinct biological entities, with hybridization identified as a threat to Trout Cod recovery and the loss of genetic identity and introgression are issues of concern in managing the existing stock structure of Murray Cod (National Murray Cod Recovery Team, 2010; Trout Cod Recovery Team, 2008). Both species have active stocking programs, with Murray Cod primarily stocked for recreational angling whilst Trout Cod are specifically stocked for conservation purposes, with an ultimate goal of returning the species to a status where it may again support a recreational fishery.

A Trout Cod conservation restocking program in the Upper Murrumbidgee released 326,200 Trout Cod fingerlings between 1988 and 2009 on 35 occasions across 8 sites (Koehn et al., 2013) with the aim of re-establishing the species in this river reach. The majority of the release sites were upstream of Geigerline Gorge (Fig. 1). However, low numbers were stocked prior to 1992 (a total of 11,000 individuals across 2 releases). The majority of fish (a total of 205,000 across 16 releases) were stocked between 2004 and 2007, with stocking ceasing in 2008. This means 24 years have elapsed since the first stocking, but only 8–10 years since the majority of the stocking occurred. Trout Cod become sexually mature after 3–5 years (Lintermans, 2007), producing noticeably more recruits after 6 years of age (Lyon, Todd & Nicol, 2012). This means that there has been opportunity for at least 4 generations to occur by 2011—the beginning of this study.

Figure 1 Upper Murrumbidgee River Maccullochella hybridisation study area.

Collection sites are shown in larger black text and putative barriers to adult fish migration are shown in smaller red text. Barriers from F Dyer, M Lintermans & P Couch (2014, unpublished data).

First generation interspecific hybridization between Murray Cod and Trout Cod has been recorded in impoundments (Harris & Dixon, 1986), hatcheries (Ho et al., 2008) and rarely in wild sympatric populations (Douglas et al., 1995). Fisherman too have reported catching hybrid fish (based on phenotypic characteristics) although not all of these anecdotal records are reliable (Cleaver, 2015) as misidentification of these two cod species was only resolved in 1972 with the formal redescription of Trout Cod (Berra & Weatherley, 1972). In any case morphological distinction between species remains difficult, particularly in larvae and juveniles.

A recent review of the national Trout Cod recovery program found that there were encouraging signs of recovery for this species, but genetic considerations were not widely canvased in the review (Koehn et al., 2013). Hybridization was identified as a major concern for one population of Trout Cod translocated to Cataract Dam, a water reservoir (Harris & Dixon, 1986; Douglas, Gooley & Ingram, 1994). This then resulted in initial stocking site selection criteria in the program excluding sites where Murray Cod were known to be present (Douglas, Gooley & Ingram, 1994). However, minimal detection of hybrids between Murray Cod and Trout Cod in riverine environments (e.g., Douglas et al., 1995) has meant that site selection criteria has been relaxed in recent years and stocking now regularly occurs where Murray Cod are present. However, if hybridization has previously occurred but not been detected, the effort to reintroduce Trout Cod may not be as effective as it might be because of inter-specific hybridization. Here we use genetic single nucleotide polymorphism (SNP) and mitochondrial sequence data to examine hybridization and introgression between Murray Cod and Trout Cod in the upper Murrumbidgee River and consider the implications for restocking programs.

Methods

Animal material

We examined 251 Maccullochella larvae which were collected in 2011, 2012, and 2013 from six sites in Murrumbidgee River in the Australian Capital Territory (ACT) (Fig. 1 and Table 1) using standard larval driftnets with 500 um mesh. For reference purposes two Trout Cod controls were included, one hatchery sourced larva obtained from NSW DPI Narrandera, the other an adult fish, from a stocked impoundment (Bendora Reservoir) in the ACT. Two Murray Cod reference samples were obtained from the upper Murrumbidgee River. Adult Maccullochella were identified based on morphological features after (Lintermans, 2007). Fish were collected under ACT Government licences LT2011516, LT2012590 and LT20133653. Research was conducted under approvals CEAE 11-15 and CEAE 13-17 from the University of Canberra Committee for Ethics in Animal Experimentation.

Larvae and tissue samples were preserved in 95% ethanol at room temperature until 2014 after which samples were stored at −20 °C. Larval fish were aged using otolith daily increments (Humphries, 2005).

Genomic DNA extraction and sequencing

Total DNA of different genotypes was isolated from whole larval heads, or for adults, from approximately 0.25 cm3 of caudal fin or muscle tissue. The DNA extraction protocol is detailed in Couch & Young (2016), and is based on a turtle DNA extraction protocol (FitzSimmons, Moritz & Moore, 1995). Each tissue sample was placed into a 1.5 mL ep tube with 300 µL extraction buffer, 30 µL SDS (20%) and 15 µL Prot K (20 mg/mL). It was incubated at 55 °C overnight, while rotating at 14rpm.

A total of 1.5 µL RNase A (4 mg/mL) was added and the mix was incubated for 30 min at 37 °C. The protein was precipitated by spinning at 13,000 rpm for one minute at room temperature. After aspirating the lysate while avoiding the cell debris pellet, the lysate was transferred to new tube for the second stage of protein precipitation. 150 µL of ammonium acetate was added and mixed. It was spun at 13,000 rpm for 30 min at room temperature and the lysate was transferred to a 1.5 mL tube, again avoiding any remaining cell debris/SDS pellet.

Table 1 Numbers of Maccullochella larvae examined per site and year.

Site name	2011	2012	2013	Total	
Tharwa sandwash	21	5	31	57	
Lanyon	6	6	10	22	
Murramore	8	5	7	20	
Kambah pool	4	2	48	54	
Bullen range	22	10	10	42	
Nerreman	3	21	32	56	
Total	64	49	138	251	

To precipitate the DNA 600 µL of isopropanol was added and mixed. The mixture was cooled at −80 °C for 5 min then at 4 °C for 30 min. It was spun for 20 min at 13,000 rpm at 4 °C. The isopropanol was decanted, leaving the DNA pellet behind. 600 µL of cold 70% ethanol was added to each tube, and gently mixed. It was spun at 13,000 rpm 4 °C for 10 min. The ethanol was aspirated and the remainder air dried while covered for 15 min. The DNA pellet was re-suspended in 40 µL filtered water.

A sub-sample of the extract was run on a 0.8% Agarose gel electrophoresis for one hour at 90v with Hyperladder 1 kb+ size standard to check the extraction quality.

Sequencing was done using DArT PL DArTseq™ which represents a combination of DArT complexity reduction methods and next generation sequencing platforms (Kilian et al., 2012; Courtois et al., 2013; Raman et al., 2014; Cruz, Kilian & Dierig, 2013). DArTseq™ represents an implementation of sequencing of complexity reduced representations (Altshuler et al., 2000) and more recent applications of this concept use next generation sequencing platforms (Baird et al., 2008; Elshire et al., 2011). Double-digest restriction associated DNA sequencing (ddRAD) is later, but similar and perhaps more widely known, implementation of genotyping-by-sequencing (GBS). The DARTseq technology is optimized for each organism and application by selecting the most appropriate complexity reduction method (both the size of the representation and the fraction of a genome selected for assays). Four methods of complexity reduction were tested in Maccullochella (data not presented) and the PstI-SphI method was selected.

DNA samples were processed in digestion/ligation reactions principally as per (Kilian et al., 2012) but replacing a single PstI-compatible adaptor with two different adaptors corresponding to two different restriction enzyme overhangs. The PstI-compatible adapter was designed to include Illumina flowcell attachment sequence, sequencing primer sequence and “staggered,” varying length barcode region, similar to the sequence reported by Elshire et al. (2011). The reverse adapter contained flowcell attachment region and SphI-compatible overhang sequence.

Only “mixed fragments” (PstI-SphI) were effectively amplified in 30 rounds of polymerase chain reaction (PCR). Amplifications consisted of an initial denaturation step of 94 °C for one minute, followed by 30 cycles of PCR with the following temperature profile: denaturation at 94 °C for 20 s, annealing at 58 °C for 30 s, and extension at 72 °C for 45 s, with an additional final extension at 72 °C for seven minutes.

After PCR equimolar amounts of amplification products from each sample of the 96-well microtiter plate were bulked and applied to c-Bot (Illumina) bridge PCR followed by sequencing on Illumina Hiseq2500. The sequencing (single read) was run for 77 cycles.

Sequences generated from each lane were processed using proprietary DArT analytical pipelines (http://www.diversityarrays.com). In the primary pipeline the fastq files were first processed to filter poor quality sequences, applying more stringent selection criteria to the barcode region compared to the rest of the sequence. In that way the assignments of the sequences to specific samples carried in the “barcode split” step were very reliable. Approximately 2,000,000 sequences per barcode/sample were identified and used in marker calling. Finally, identical sequences were collapsed into “fastqcoll files.” The fastqcoll files were “groomed” using DArT PL’s proprietary algorithm (http://www.diversityarrays.com) which corrects a low quality base from a singleton tag into a correct base using collapsed tags with multiple members as a template. The “groomed” fastqcoll files were used in the secondary pipeline for DArT PL’s proprietary SNP (presence/absence of restriction fragments in representation) calling algorithms (DArTsoft14). For SNP calling, all tags from all libraries included in theDArTsoft14 analysis are clustered using DArT PL’s C++ algorithm at the threshold distance of 3, followed by parsing of the clusters into separate SNP loci using a range of technical parameters, especially the balance of read counts for the allelic pairs. Additional selection criteria were added to the algorithm based on analysis of approximately 1,000 controlled cross populations. Testing a range of tag count parameters facilitated selection of true allelic variants from paralogous sequences, In addition, multiple samples were processed from DNA to allelic calls as technical replicates and scoring consistency was used as the main selection criteria for high quality/low error rate markers. Calling quality was assured by high average read depth per locus (>60).

Marker scoring and statistical analysis

DArTsoft (Diversity Arrays Technology, Building 3, University of Canberra, Australia), a software package developed by DArT PL (http://www.diversityarrays.com/software.html), was used to both identify and score the markers that were polymorphic. The results of polymorphic scoring are presented in Microsoft Excel in binary format with two lines per individual, the first line nominated as a reference snp marker and the second as the alternate. In each “1” denotes the presence and “0” the absence of a marker in genomic representation of a sample.

Mitochondrial DNA sequencing

To identify female parent species, and to potentially verify hybridization identified by SNP analysis we amplified a portion of the cytochrome b (cytb) gene via PCR. Samples were amplified via nested PCR due to the low quantity of DNA present. In the first reaction we used the primers Glu18 TAACCAGGACTAATGRCTTGAA and hd.macc.632 GATTTTATCTGAATCTGAGTTTA followed by Glu31 TGRCTTGAAAAACCACCGTTGT and hd.Mac.538 GGGAAGAGGAAGTGGAAGGC in the second reaction.

The first reaction used 1 µL of template DNA, 0.5 µL of each primer, 5 µL of Bioline MyTaq Red Mix and 3 µL of water 9.5 µL in total. Amplification parameters were as follows: 94 °C for 2 min followed by 35 cycles of 94 °C for 20 s, 48 °C for 20 s, and 72 °C for 60 s, and 72 °C for 7 min. This first PCR reaction was then diluted to 1:49 and 1 µL from that was used in the second 25 µL reaction with the same PCR conditions listed above. We examined PCR products on a 2% agarose gel using SYBR safe DNA gel stain (Invitrogen, Eugene, OR, USA). Sequences were obtained via cycle sequencing with Big Dye 3.0 dye terminator ready reaction kits using 1/16th reaction size (Applied Biosystems, Foster City, CA, USA).

Sequencing reactions were run with an annealing temperature of 52 °C and following the manufacturer’s protocol. Sequenced products were purified using sephadex columns. Sequences were obtained using an Applied Biosystems 3730 XL automated sequencer at the Brigham Young University DNA Sequencing Center. All sequences obtained in this study were deposited in GenBank, accession numbers KX355263 –KX355274.

Analysis of mitochondrial DNA sequence data

Sequences were edited using Chromas Lite 2.0 (Technelysium, 0000) and imported into BioEdit 7.0.5.2 (Hall, 1999). Sequences coding for amino acids were aligned by eye and checked via amino acid coding in MEGA 6.06 (Tamura et al., 2013) to test for unexpected frame shift errors or stop codons.

SNP analyses

The polymorphisms identified by DArTseqs and selected for genotyping were all SNPs. The unique Maccullochella loci identified were reduced by filtering out those with call rate less than 0.98 and with a reproducibility score of less than 1. The data set used for population analysis from the DArTsoft14 pipeline consisted of 69 bp fragments containing one or more SNPs. Samples were in some cases quite degraded. It is notable that this short sequence technique was still able to reliably produce sequences with this material.

Variation in the genome-wide SNP data of the studied Maccullochella genotypes was analysed using Discriminant Analysis of Principal Components (DAPC) using sequential K-means and model selection to infer genetic clusters (Jombart, Devillard & Balloux, 2010) using R package ‘adegenet’ version 2.0.1 (Jombart, 2008). The data were converted into a genlight object and three principal components were retained. Two principal components were plotted using ggplot2 version 2.1 (Wickham, 2009). Summary and comparative statistics as well as percentages, chi squared and t-tests were created in R version 3.3.0 (R Development Core Team, 2014) and Tableau version 9.2 (Tableau, 2013). Maps were created using ARCGIS version 10 (Esri, 2013) and Tableau.

Assignment of hybrid status

Larva assignments as pure, F1 or backcrosses were derived following consideration of the assignments from clustering, DAPC of SNPs and mitochondrial sequences.

The hybrid status of larvae was assessed, initially by K-means clustering, and then with NewHybrids version 1.1 (Anderson & Thompson, 2002). NewHybrids, using the framework of Bayesian model-based clustering, computes by Markov chain Monte Carlo, the posterior probability of an individual belonging to each of the predefined distinct parental or hybrid classes, i.e., pure, F1, F2, or backcrosses. The entire SNP matrix was too large to analyse in NewHybrids, so we randomly selected 200 polymorphic loci in ten separate sets and analysed each set in NewHybrids. In this analysis, six genotype frequency classes were used; pure Trout Cod, pure Murray cod, F1, F2, Trout Cod backcross and Murray cod backcross (Table 2). Independent runs were initiated from different starting points and the MCMC chain was allowed to run for at least 100,000 iterations after burn-in until the log likelihood values reached stationarity and posterior probabilities of assignment to a class did not vary. This was repeated for each of the sets. Log likelihood values reached stationarity very quickly (c. 1,000 iterations), but a burn-in of at least 13,000 iterations was allowed before recording assignment probabilities. As a result individuals were each assigned ten probabilities of being within each of the six genotype frequency classes. The mean and standard deviation of the ten probabilities for each fish was then calculated. Known Trout Cod and Murray Cod samples were nominated a priori as parental taxa as they are the only two Maccullochella species found in the upper Murrumbidgee River within the Murray Darling Basin. Both were included in the NewHybrids mixture.

Table 2 Expected frequency of loci from parental Maccullochella species and hybrid offspring.

Expected frequency of loci used by r package NewHybrids to assign individuals to a genotype class.

Name of genotype frequency class	Expected frequency of loci from species TC or MC	
	TC	TC MC	MC TC	MC	
Pure Trout Cod (TC)	1.0000	0.0000	0.0000	0.0000	
Pure Murray Cod (MC)	0.0000	0.0000	0.0000	1.0000	
F1	0.0000	0.5000	0.5000	0.0000	
F2	0.2500	0.2500	0.2500	0.2500	
Trout Cod Backcross	0.5000	0.2500	0.2500	0.0000	
Murray Cod Backcross	0.0000	0.2500	0.2500	0.5000	

To support hybridisation findings and identify the directionality of that hybridisation, fragments of mitochondrial DNA were sequenced from two Trout Cod reference samples, two Murray Cod reference samples, and each of the F1 and backcross hybrids detected using DArT sequencing and NewHybrids.

Results

Maccullochella hybrids and control larvae species assignments

The number of unique Maccullochella SNPs analysed in the DaRT sequences was 12,728, each with 2 alleles. Of these 6,364 loci, those with a call rate above 0.98 and with a reproducibility score of 1 were selected for further analysis.

The majority of the 251 larvae genotyped are Murray Cod (239), with two known Trout Cod controls and 8 hybrids. The percentage of hybrid larvae varied from 2.1–6.1% in each year. K-means clustering identified three groups; Murray Cod, Trout Cod and hybrids. The same groups of species and hybrids can be seen clearly separated in the DAPC plot, with F1 hybrids and backcross hybrids being placed intermediate between the two species (Fig. 2). No pure Trout Cod larvae were detected.

Figure 2 Principle Components of Maccullochella snps.

PCA plot of Maccullochella snps from the upper Murrumbidgee River 2011–2013 (n = 278: 251 larvae, 25 adults, two known Trout cod). Trout Cod samples are in light blue on the right, Murray Cod in purple at the origin. The first and second generation hybrids (red and brown respectively) can be seen in between. The two third generation larva (fish #102 and #106) are in green and #106, is partially obscured within the purple Murray cod points. The two points with ♀MC had a Murray Cod as female parent based on their mtDNA. All other hybrids larvae had a Trout Cod as the female parent lineage.

The program newHybrids assigned a probability of each of the samples being in each genealogical class in accordance to the expectation from exploratory K-means clustering and DAPC. The probability assignment of each of the ten replicates with a randomly selected set of 200 snps were identical (Table 3), thus SD = 0, apart from fish #102 which had a mean probability of 0.8 (SD = 0.42) of being a Murray Cod and a mean probability of 0.2 (SD = 0.42) of being a Murray Cod backcross. The mitochondrial sequences were identical for the eight Trout Cod and six of the eight hybrid sequences tested, similarly the four homologous Murray Cod sequences of the same length also showed no nucleotide diversity. The Trout Cod and Murray Cod mitochondrial sequences differed from each other at 73 of the 537 loci in the sequence. Mitochondrial sequencing indicates that six of the eight hybrid larvae had a Trout Cod as a female parent and female grandparent, while one first generation backcross larva (fish #262) and one second generation backcrossed hybrid (fish #102) had a female Murray Cod parent (Table 3).

Table 3 Hybrid and reference Maccullochella species and hybrid genotype class assignments.

DArT genome reduction analysis by k-means clustering (12299 DArT snps), NewHybrids assignments based on 10 runs of 200 randomly selected SNPS, and matrilineal assignments from mitochondrial sequencing. Shaded lines are reference samples. Parental Maccullochella peelii (n = 25) samples are not shown.

Fish	Clustering indication	NewHybrids assignment	NewHybrids mean (p) (ten runs of 200 snps)	Mitochondrial DNA (♀)	Assignment based on NewHybrids and mitochondrial	
76	MC	Pure MC	1 (SD = 0)	MC	Pure MC	
100	MC	Pure MC	1 (SD = 0)	MC	Pure MC	
102	MC	Pure MC MC Backcross	0.8 (SD = 0.42) 0.2 (SD = 0.42)	MC	MC BX × MC	
106	MC	Pure MC	1 (SD = 0)	TC	MC BX × MC	
141	Hybrid	F1	1 (SD = 0)	TC	F1	
145	Hybrid	F1	1 (SD = 0)	TC	F1	
178	Hybrid	MC Backcross	1 (SD = 0)	TC	MC Backcross	
262	Hybrid	MC Backcross	1 (SD = 0)	MC	MC Backcross	
269	Hybrid	F1	1 (SD = 0)	TC	F1	
302	Hybrid	MC Backcross	1 (SD = 0)	TC	MC Backcross	
Bend	TC	Pure TC	1 (SD = 0)	TC	Pure TC	
Narra	TC	Pure TC	1 (SD = 0)	TC	Pure TC	

The hybrids in Table 3 represent the product of 13 inferred matings; six first generation crosses, six second generation crosses and two third generation cross (fish #102 and #106). Of these 13, at least eight involved a female Trout Cod. This is a statistically significant departure from the 50% expected if matings were random, with a chi squared value of χ2 = 4.76 (df = 1, p = 0.03).

Location and temporal aspects of hybrid larvae

Hybrid larvae were detected at four of six sites sampled; Tharwa, Lanyon, Murramore and Nerreman (Fig. 1). There were three hybrid larvae detected in the 138 larvae caught and sequenced in 2013 (2.17%). This included one F1 hybrid and two backcross hybrids. There were three hybrids sampled of 49 larvae sampled in the previous year 2012 (6.12%) This included one F1 hybrid and two backcross hybrids. Two backcross hybrids were detected in the 64 larvae sampled in 2011 (3.13%).

There was no significant difference between the day of the year on which hybrid and non-hybrid larvae were sampled (t =  − 0.162, df = 10.415, p-value = 0.874). There was no significant difference between the age of hybrid and non-hybrid larvae sampled (t =  − 0.053, df = 7.12, p-value = 0.959).

Discussion

Hybridization and genetic effects

This is the first study to confirm the occurrence of hybrid Trout Cod–Murray Cod in the Upper Murrumbidgee River, and the first record of hybrid larvae in the wild. Although hybrid larvae, juveniles and adults have been recorded in hatcheries and impoundments, and rare hybrid adults have been recorded in rivers previously (Douglas, Gooley & Ingram, 1994; Douglas et al., 1995), this is the first time fertile first generation (F1) hybrids have been recorded in the wild as evidenced by the finding of F1 × Murray Cod backcrosses (F1 × MC). These F1 × MC backcrosses have also been shown to be fertile as there are two examples of F1×MC backcross again backcrossing with a Murray Cod (fish #106 and #102). All backcrosses were with pure Murray Cod. Such introgression has not been recorded previously in these two species in riverine populations.

The assignment of genotype genealogical class (parental, F1, backcross, etc.) based on the genotype frequency class from newHybrids, as noted by Anderson & Thompson (2002) and Fitzpatrick (2012) applies only to the first two generations of interbreeding. Other generational mixes or deeper generations become indistinguishable using this method which is why a specific generational depth cannot confidently be assigned to fish #102 or #106, although we can say it is a 3rd or greater generation.

This is also the first time stocked riverine Trout Cod have been confirmed as reproducing in the upper Murrumbidgee, but notably no pure trout cod larvae were detected. None of the backcrosses were with a pure Trout Cod. While successful breeding of the first born generation has been used by some as a measure of success of reintroductions (Sarrazin & Barbault, 1996), breeding alone is not a useful measure of success (Lintermans et al., 2015). Such breeding, unless genetically sound, and sustained, is a measure of re-introduction success rather than a more important indicator of recovery success. This study suggests sympatric restocking of endangered Trout Cod poses concerns from hybridization and is an important consideration for Murray Cod and Trout Cod management.

The mitonuclear discordance observed here is unlikely to be due to historic interbreeding between the two naturally occurring sympatric species rather than the recent reintroductions. Trout cod have been absent from the ACT reach of the upper Murrumbidgee River system for more than 20 years prior to reintroductions commencing. Also, the hybrid Trout cod mitochondrial sequences were identical to the originating hatchery mitochondrial sequences. This too suggests that they originated from the stocked fishes rather than the original wild type. All stocked trout cod in Australia have come from broodstock originating from the one remaining wild trout cod population in the Murray River below Yarrawonga, Victoria.

There are historic reports that Trout Cod spawned earlier than Murray Cod (Cadwallader, 1977) but more recent research shows that Trout Cod have a shorter, but overlapping spawning period with Murray Cod (Koehn & Harrington, 2006). In this study sampling commenced 51, 42 and 24 days in 2011, 2012, 2013 respectively, before the first Maccullochella larvae were detected dispersing. This, and the finding that hybrids hatching did not differ significantly from Murray Cod hatching suggests that it was not a temporal sampling issue that resulted in no pure Trout Cod larvae being detected and we would have detected them if they were present. At the present time it is unclear what evolutionary outcome is most likely from the hybridization observed in this study. The relative fitness of hybrid Maccullochella is unknown. It is possible that reduced fitness of larval hybrids means they rarely survive to adulthood, and so the implications of hybridization are minimal for the conservation of riverine populations. Alternatively, hybrid vigour (heterosis) may be evident, with hybrids demonstrating enhanced fitness and potentially leading to emergence of a hybrid swarm. There is literature that highlights reduced fitness in hybrid fish (see for example Houde, Fraser & Hutchings (2010)). However, there are also some important examples of heterosis in a number of fish species. Salmonid heterosis for resistance to amoebic gill disease is one Australian commercial fish breeding example (Maynard et al., 2016). In any case, outbreeding depression is likely and longitudinal observations of a variety of age classes will be required to determine the outcome.

Seehausen (2006) provides forewarning that homogenizing environments may cause the rapid loss of species through a reversal of the speciation process. One clear example of two species becoming one in freshwater fishes is the lacustrine stickleback study undertaken by Taylor et al. (2006). Such an outcome in two threatened species such as Trout Cod and Murray Cod is highly undesirable.

Dispersal and the limited male hypothesis

The absence of pure Trout Cod larvae and the relatively high levels of hybrid larvae detected, given the limited number of Trout Cod expected to have matured following restocking, raises an important question as to why this hybridization is occurring. It is considered unlikely that the mis-matings are a result of hatchery-induced changes in behavior of Trout Cod. All trout Cod were stocked as fingerlings (35–55 mm total length) derived from wild caught broodstock (Koehn et al., 2013). Previous work has shown pronounced differences in survival and movement between hatchery derived adult fish depending on whether they were ongrown for an extended period or stocked as fingerlings (Ebner et al., 2006; Ebner, Thiem & Lintermans, 2007; Ebner & Thiem, 2009). They suggest that prolonged hatchery rearing is the more likely cause of altered behavior. Given that male Trout Cod are more limited in abundance than male Murray Cod, one hypothesis is that this could result in a limited number of mature stocked female trout cod succumbing to a disproportionately high mating pressure from more numerous Murray Cod males, rather than locating scarcer Trout Cod males. Both cod species have a similar reproductive strategy and overlapping spawning season (Koehn & Harrington, 2006; Lintermans, 2007) as demonstrated by hybridization in both lentic and lotic environments (Douglas, Gooley & Ingram, 1994; Douglas et al., 1995). Consequently, reproductively ripe individuals of both species are likely to be present in the river at the same time. Mitochondrial sequencing of the hybrid larvae in this study supports this ‘limited male’ Trout Cod hypothesis, but not exclusively as the female parent of two hybrid larvae (fish #102 and #262) were found to be a Murray Cod. Collection and testing of a larger number of larvae will provide a better estimate of the bias towards Trout Cod as the female parent of hybrids.

Dispersal of post-juvenile Trout Cod away from stocking sites has been previously postulated as one explanation for the low detectability of Trout Cod in subsequent monitoring programs (Ebner, Thiem & Lintermans, 2007; Ebner et al., 2006; Ebner & Thiem, 2009). Such dispersal may also contribute to low density of adult fish, and subsequent increased pressure to mate with more abundant congeners. The ‘limited male’ hypothesis might also be exacerbated by skewed sex ratios resulting from restocking programs. At least one study has found deviation from the expected sex ratio where females dominated by 2.5 to 1. The same authors report previous unpublished findings of highly skewed sex ratios of up to nine males to each female (Lyon, Todd & Nicol, 2012). Identification of sex-linked markers would be helpful for future studies of this and other species.

In the upper Murrumbidgee River Murray Cod had a limited distribution, with the species not recorded in reaches upstream of a barrier formed by Gigerline Gorge (Fig. 1) when the Trout Cod stocking program commenced in 1988 (Lintermans, 2002). Murray Cod are known to undertake upstream spawning migrations (Koehn et al., 2009) but adult Trout Cod are less mobile, at least in lowland rivers, than Murray Cod (Koehn & Nicol, 2016). The major stocking site for Trout Cod (99,500 fish from 1996 to 2005) was at Angle Crossing immediately upstream of Gigerline Gorge and so the presence of a migration barrier may result in aggregations of reproductively ripe Murray Cod mixing with downstream displaced trout cod below the barrier, further enhancing the chance of hybridization. Although 99,500 Trout Cod fingerlings was a substantial stocking effort over a 10 year period, the relatively high fecundity of the species means that this stocking effort only represents what would be the naturally expected reproductive output of fewer than 20 individuals based on the egg and larvae mortality estimates of Ingram & Rimmer (1993) and Todd, Nicol & Koehn (2004). The majority of the hybrids were detected less than 10 km downstream of Gigerline Gorge (Fig. 1), with this location having one of the last naturally occurring remnant populations of Trout Cod prior to their extirpation in the late 1970s (Berra, 1974; Lintermans, Kukolic & Rutzou, 1988).

Implications for restocking

Potential implications of genetic effects resulting from restocking have been highlighted for some time, even when there was a greater paucity of data about the genetic structure of fish in the MDB (Phillips, 2003; Gillanders, Elsdon & Munro, 2006). The findings in this study are a specific case of genetic effects resulting from stocking programs. Rourke et al. (2010) have previously noted a range of genetic effects from stocking. The introgression observed in this study, although clearly resulting from a restocking program, cannot be meaningfully compared to the expected genetic effects of the two species coexisting naturally because, although they were sympatric before extirpation of Trout Cod in the upper Murrumbidgee in the 1970s, the relevant abundance and demographic data does not exist. However, if the limited numbers of mature female Trout Cod resulting from stocking are under a disproportionately high mating pressure from Murray Cod males compared to when high number of both species naturally coexisted, then there is likely to be proportionately more hybridization than may have occurred previously. If so, this is a genetic effect that should be given attention and considered when making conservation restocking decisions in these and other species.

Although hybridization is a natural process and is relatively more common in fishes than other vertebrates, the occurrence of hybridization and introgression poses some real challenges for threatened species recovery programs (Gese et al., 2015). Reintroductions of threatened fish are usually resource limited, and so the number of individuals available from captive breeding programs is often only equivalent to the reproductive output of a handful of wild spawnings. Consequently, when trying to establish wild populations in the presence of an abundant congener, mis-mating is highly possible. This is in contrast to genetic swamping when a large number of hatchery-bred fish are stocked over the top of a small remnant population, as has occurred with Eastern freshwater cod (Nock et al., 2011).

The national reintroduction program for Trout Cod originally used several criteria for selecting stocking sites, including one criterion that stocking should not occur where Murray Cod was present (Douglas, Gooley & Ingram, 1994). This was in recognition of the possibility of hybridization (as previously demonstrated in Cataract Reservoir Wajon, 1983; Harris & Dixon, 1986) and was an important consideration when selecting Trout Cod stocking locations in the upper Murrumbidgee River, with all main stem stockings prior to 2005 occurring upstream of Gigerline Gorge where Murray Cod were considered absent (Lintermans, 2002). In subsequent iterations of the stocking program, this criterion was discarded, and most stocking locations now have wild populations of Murray Cod present. Perhaps unfortunately, fishing clubs, with the assistance of NSW Fisheries instituted a stocking program from 2008–2011, in a number of tributaries in the reach upstream of Gigerline Gorge of more than 4,000 Murray Cod for recreational purposes (Cooma-Monaro Express, 2015). Since these stockings commenced, Murray Cod larvae have now been collected upstream of Gigerline Gorge, indicating that Murray Cod are now breeding in this reach. So this reach now contains low abundances of both species, possibly leading to limited mating opportunities, and raising the potential for mis-mating.

The earliest introgression detected in this study is an F1 × MC backcross × Murray Cod larva (fish #102 and #106) which indicates at least three hybrid generations by 2011. This suggests the first F1 hybrid mating in this lineage took place between 1998 and 2002. Given restocking commenced in 1988 and increased after 1992, introgression in even deeper backcrosses is possible but more sampling and species specific sensitive analytical techniques might be used to identify evidence of this.

The national Trout Cod restocking program has been through a number of iterations and changes in approach, with stocking moving from releases of small numbers of fish (<1,000) for one or two years to releases of tens of thousands of fish for 5–10 years (Lyon, Todd & Nicol, 2012; Koehn et al., 2013). The upper Murrumbidgee stocking program sits midway in this stocking approach and likely still suffered from insufficient fish being stocked over a concentrated temporal and spatial scale. The release of 99,500 fish over 10 years at the stocking site immediately upstream of Gigerline Gorge was the exception rather than the rule, with subsequent stocking efforts in the upper Murrumbidgee releasing fish over much shorter timespans (average annual release per site of 10,900; sites stocked for 1 to 5 years) (NSW Fisheries, 2010, unpublished data). This is substantially less than the most recent successful reintroduction of this species which released 277,460 fish over 10 consecutive years (Lyon, Todd & Nicol, 2012). Upstream of the Gigerline Gorge, stocking low numbers was probably not a major issue as Murray Cod were not present, and so hybridization and introgression could not occur there. However this is no longer the case, mature individuals of both species are now obviously present. If the upper Murrumbidgee Trout Cod restocking program is to be successful and minimize the chances of hybridization and introgression with Murray Cod, then stocking even greater numbers of Trout Cod over extended timeframes (∼10 years) maybe required.

Conclusion

Given the single annual spawning reproductive strategy of the Trout Cod, each hybridization event is a precious but wasted reproduction opportunity for this species, which is listed as endangered under the Australian Environmental Protection and Biodiversity Conservation Act (Department of Environment, 2016). The National Recovery Plan for the Trout Cod, Maccullochella macquariensis (Trout Cod Recovery Team, 2008) noted potential risks of hybridization but limited recommendation on the matter to ‘…caution should be exercised in stocking Murray Cod in the same waters.’ The present study clearly demonstrates hybridization and introgression between these species, highlighting that even greater caution should be exercised when stocking Murray Cod into waters where a Trout Cod recovery program is extant but Murray Cod are not.

We thank Matt Young for his invaluable assistance in the lab, particularly with DNA extraction and PCR. We also thank Alica Tschierschke for much technical assistance in the lab and proficiency with ArcGIS. For adult Murray Cod DNA extraction and mitochondrial sequencing we thank Paul Sunnucks, Sasha Pavlova and team. We thank the editor and reviewers for their generous comments on the manuscript. We also are grateful to NSW Department of Primary Industries Hatchery at Narrandera, NSW who provided Trout Cod larval samples.

Additional Information and Declarations

Competing Interests

Author Contributions

Animal Ethics

Field Study Permissions

DNA Deposition

Data Availability

The authors declare there are no competing interests.

Alan J. Couch conceived and designed the experiments, performed the experiments, analyzed the data, wrote the paper, prepared figures and/or tables, reviewed drafts of the paper.

Peter J. Unmack conceived and designed the experiments, performed the experiments, contributed reagents/materials/analysis tools, reviewed drafts of the paper.

Fiona J. Dyer and Mark Lintermans conceived and designed the experiments, reviewed drafts of the paper.

The following information was supplied relating to ethical approvals (i.e., approving body and any reference numbers):

University of Canberra Committee for Ethics in Animal Experimentation: Research was conducted under approvals CEAE 11-15 and CEAE 13-17.

The following information was supplied relating to field study approvals (i.e., approving body and any reference numbers):

Fish were collected under ACT Government licences LT2011516, LT2012590 and LT20133653.

The following information was supplied regarding the deposition of DNA sequences:

Genbank accession numbers: KX355263–KX355274

The following information was supplied regarding data availability:

The data are available as a genlight object on the Github repository (https://github.com/dnatheist/Maccullochella-hybridisation), along with the code required to reproduce the DAPC outputs and a text file (snpsHybridDataRaw.csv) with raw comma delimited SNP data prior to filtering.

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
