# Peer review of "Who’s your mama? Riverine hybridisation of threatened freshwater Trout Cod and Murray Cod"

_PeerJ, doi:10.7717/peerj.2593_

## Round 0.1 · original submission · Major Revisions

We have received three detailed reviews regarding your manuscript. While reviewers were generally positives about the study, they also raised a number of substantial issues that will deserve further revisions. In particular, the introduction and discussion were deemed confusing and insufficiently focused. Also, the presentation of general context of the study and objectives should be improved. There is also a lack of details concerning some of the analyses presented and a confusion regarding information that should be included in the Results and the M&M sections respectively. Finally, some of the tables and figures also need to be modified. The comments of the reviewers concerning all the points mentioned above are very well detailed and all their other comments should also be taken into account when revising this manuscript.

·

Basic reporting

I have now been throughout the revision of the manuscript untitled "Who’s your mama? Riverine hybridisation of threatened freshwater Trout Cod and Murray Cod." written by Couch and collaborators.
While the manuscript in appearance conforms the standards, I am afraid that for most of the analyses there is a confusion between the Results and the M&M section. Some parts that should be provided in Results already appear in M&M and vice-versa. And some tests appear in the Results section thought they were not even mentioned in M&M.
Please see my all detailed comments on the manuscript with the attached document I provide here.

Experimental design

The authors described well the research question, and developed an appropriate sampling design to answer the question, respecting the ethics in application inside their country.
I also like the fact that the authors combined the NGS approach to the Mitochondrial sequencing, really appropriate here.
However I am quite concerned by the fact that (i) the authors omitted quite often to provide details about their analysis, like parameters they actually used for running an analysis and (ii) the authors are confused and elusive about their DArTseq approach (both for M&M and Results section) and are using a vocabulary not conform to the standard of NGS analysis. At least for a better understanding they should provide more details to help people not familiar with DArTseq approach. In this way, I provide all along the manuscript comments that would help authors to make to clearer. Concerning, both mitochondrial and DArT data it would be really great to see more results than that, at least to have an appreciation of the the level of polymorphism (Heterozygosity indices, nucleotide diversity, numbers of haplotypes defined with mitochondrial data)

Validity of the findings

Finally, I really think the findings of this paper are correct and that good deployement have been made to make them valid and to answer the research question but for a better assessment of the validity of the findings, I really need to get insights from authors (as mention in the previous box).

Additional comments

So far, I encourage the authors to follow-up my detailed comments I have made all along the ms in the attached document. Hope it would help them to improve their ms before a re-submission.
I also think because the authors chose to publish their data in a PeerJournal they should demonstrate more transparency, detailing their analysis but also providing accessible dataset, for example the DArtseq binary file output. That would really help people not familiar with DArT output to get more confident with the study.

Sincerely

Reviewer 2 ·

Basic reporting

Overall, this was a well-executed, straightforward study of hybridization with practical applications for the two species in question and reintroduction/stocking efforts for either one. For their revisions, I would suggest the authors particularly focus on honing the writing throughout, especially in the introduction and discussion, and improving the quality of Tables and Figures (formatting such as avoiding 'plain boxes', content, larger axis font size, etc.). The general conceptual context of the research study and specific study objectives could be substantially improved; listed below are a number of suggestions for doing so.

Specific comments
Abstract – Unless specified by the journal, avoid having references in the abstract (e.g. Allendorf et al. 2001). It only adds unnecessary word count and breaks the flow of the results reporting and context of the study.
Introduction – lines 38-62. The text here is useful and important to the paper’s context, but it could be improved, specifically more balanced and more focused. Particularly with respect to hatchery-reared fish and ‘restocking’ (or ‘stocking’, or ‘supplementation’), the focus of the referencing is more on the purported negative effects of hatchery-wild mixing and introgression between populations, and not enough on the benefits ecological, genetic or otherwise, nor is the weight of evidence for when hybridization is detrimental vs. beneficial in a conservation context fully conceptualized. In particular, the focus of the genetic concerns is on ‘neutral’ population genetic diversity (allelic richness and heterozygosity) only, not quantitative genetic variation and how adaptive genetic variation might (or might be) be impacted. This is a messier subject to handle for sure, but more recent literature could be surveyed to flesh out a more comprehensive introduction (e.g. see A. Harbicht et al. 2014, Evolutionary Applications; F. Lamaze et al. 2014, Molecular Ecology; both papers are on brook trout/charr); it seems pertinent because undoubtedly hybridization between Murray and Trout Cod will cause changes in human-desired phenotypic attributes such as body size.
Also, I am not sure how the information on lines 38-62 is being ‘funnelled down’ to the goals and context of the present study. Reading further along on lines 63-96, is the main issue inter-specific hybridization between Murray Cod and Trout Cod, or is it the possible loss of intraspecific genetic diversity in Trout Cod from hybridization with a hatchery source (or the same in Murray cod)?
Finally, the objective of the study on lines 106 -108 is quite coarse: how specifically is hybridization being assessed in these two species (what is the general analytical approach), what is a specific implication for management if the two species are in fact hybridizing naturally?

Experimental design

In general, the experimental design for assessing hybridization between the two species of cod is solid. More discussion on how multi-generational hybrids were deciphered with NewHybrids software would be useful, and consider randomizing subsamples of SNPs for analyses.

Materials and Methods
Lines 116-117: how many Murray Cod controls were screened? Provide specific numbers.
Line 190-191, 195-197: the calling quality is reported as an average read depth per locus. Can the authors provide additional information on individual sample quality, e.g. individual – level call rate? And what was the threshold used to keep or reject individuals from further analyses?
Lines 199-200. Provide a brief explanation for why mtDNA is being screened, it may not be obvious for some readers.
Lines 223-225. Report the number of SNPs in the final dataset here.
Lines 235-240. Instead, why not randomly subsample 200 SNPs from the whole dataset, and repeat the procedure 10 or preferably 100 times, to generate confidence intervals of admixture per individual? Picking the most polymorphic loci is potentially unrepresentative of true values of genome-wide hybridization.
General comment: There needs to be more description of how NewHybrids discerns F1, BC and F2 or later generation hybrids from one another. This is pertinent for the Results section when discerning exactly when hybridization took place, and whether backcrosses are F2 generation mixes or go back even further (up to 4 generations).

Validity of the findings

Reporting and interpretation of the results is sound; formatting of some tables and figures could be improved (see below).
Lines 253-254. This explanation would be better placed in the M&M (see above comment)

Additional comments

Discussion
Lines 285-289. Again, it is difficult to interpret whether Trout Cod hybridizing with Murray Cod is a reintroduction ‘success’ because success and the goals of ‘restocking’ were not clarified sufficiently in the introduction. If the goal of restocking is to reintroduce Trout Cod into this river where it was previously extirpated, then the authors provide evidence that the program has not been successful because pure Trout Cod are not being produced over successive generations following stocking. If the goal is to supplement Murray Cod with outside genetic material, then perhaps the restocking is achieving this. And if this assessment is more to do with investigating the potential concern of hybridizing between two taxonomically defined species, then clearly restocking can be an issue for Murray Cod and Trout Cod management.
Lines 303-304. ‘Most of the literature…..’ is a pretty bold statement that needs to be toned down a bit because there have been no rigorous quantitative syntheses to assert this point. For sure, there is a general focus on the negative aspects of hybridization between hatchery or farmed and wild fish, but there are also numerous examples of hybrid fish doing very well.
Lines 303-312. Some comment on the demographic context of stocking would be useful here. Namely, the rate or extent of stocking might simply be low relative to the number of Murray Cod larvae in the river, reducing the chance of high levels of hybridization between the species, irrespective of the evolutionary consequences of said hybridization.
Lines 315-317. Absolutely! So the big question is, why bother stocking Trout Cod into this river at all? Was this just a case where genetic consequences of stocking were overlooked before the fish were dumped in?
Lines 338-341. What would cause the deviation in sex ratio of stocked fish? Is there sexual-size dimorphism in this species, i.e. would the faster growing sex be favoured?
Table 1 could be enhanced for publication, in terms of style.
Table 2. can probabilities include confidence intervals?
Research suggestion. The authors have three consecutive cohorts of larvae with reasonable sample sizes, and the vast majority originate from one species, Murray Cod. The genetic data for these cohorts could be used to infer the effective number of breeders (Nb) which generated each cohort, using the LDNe method. Presumably Murray Cod are iteroparous; Nb is an analogue of Ne in iteroparous species (Waples et al. 2013, PRSL), but more importantly, the estimate of Nb per year could be used to approximate how many breeding adults (N) existing in the river each year, based on known Nb/N ratios in species with related life histories. This might provide a nice complement to the concern of hybridization from stocking of Trout Cod in the drainage, if it turns out that the Murray Cod population has a low number breeders, and Nb estimates might also help to supplement some of the authors ideas about asymmetric mating and limited males.

Reviewer 3 ·

Basic reporting

• Structure conforms to PeerJ standard.
• Mostly clear, unambiguous, professional English is used throughout.
• Introduction and background do NOT clearly show context –see additional comments.
• Literature is mostly well referenced and relevant, but references require editing in text and reference list.
• Figures are relevant, high quality, well labelled and described.
• Raw data has NOT been supplied.

Experimental design

• Original primary research, within the scope of the journal.
• Research question is fairly well defined, relevant, and meaningful.
• Rigorous investigation has been performed to a high technical and ethical standard.
• Methods are NOT described with sufficient detail & information to completely replicate – in particular regarding the filters used to select reliable SNPs. However, details are sufficient that a researcher could probably apply their own SNP quality control pipeline and identify the same hybrid individuals.

Validity of the findings

• Data is robust, statistically sound and controlled.
• Conclusions are fairly well stated and linked to original research question and supporting results.
• Speculation is identified as such, although some management suggestions are made on the basis of this speculation.

Additional comments

This manuscript presents the results of a study that uses SNPs (genotyped via a RAD sequencing approach) to investigate the reproductive success of stocked Trout Cod (Maccullochella macquariensis) in the Murrimbidgee River of south-eastern Australia, which also supports the closely related Murray Cod (M. peelii). The authors netted larval Maccullochella larvae from multiple sites over multiple years and genotyped them (together with Trout Cod and Murray Cod reference samples) for 12,299 SNPs. They found no pure Trout Cod larvae, however approximately 3% of larvae were Trout Cod and Murray Cod hybrids (mainly F1 and Murray Cod backcross). The direction of the matings that generated these hybrids was further investigating by genotyping mtDNA. The authors discuss their results with respect to the Trout Cod stocking program.

The methodological approach is fine: collection of samples was sufficient, the genotyping approach is robust (although there is insufficient information currently provided about the SNP calling pipeline to enable it to be exactly replicated), and analyses appear appropriately applied. However, I found the introduction and some of the discussion confusing and insufficiently focused. I therefore feel that the manuscript requires revision before it can be accepted for publication.

Detailed comments are as follows:

In particular, the introduction and parts of the discussion read as though the authors have plucked sentences from a conservation genetics textbook rather than having a true understanding of the population genetic issues involved. They do not even mention outbreeding depression, which is the main conservation concern where novel introgressive hybridization is occurring between genetically distinct taxa. Further, in places their discussion conflates the potential genetic effects of stocking programs on the stocked taxon and the potential effects of introgressive hybridization from the stocked population into a different taxon – two different (although inter-related) issues.

Lines 39-42: the authors start by noting that hybridization and introgression play important evolutionary roles. This is true, but addressing these evolutionary roles is not the aim of this manuscript. It would be more relevant to start off here with a clear introduction about the conservation genetic implications of introgressive hybridization between taxa.
Line 44 ‘determining whether hybridization is beneficial or detrimental for the species involved is commensurately important for conservation’ – this might be true (although it is very difficult), but it is not what the authors are doing in this study – they are merely documenting the presence of hybrid individuals.
Line 47: this line effectively repeats what the authors have already said in line 39.
Line 48: ‘introgression… can eventually reduce genetic diversity…’ – how does it do this?
Line 51: ‘the synergistic effects of multiple extinction drivers … are only starting to be understood’ – this is true but again, this is not what is being addressed in this paper.
Line 52: ‘the genetic consequences of introgression are of increasing interest …. as many species are on an irreversible path to extinction (the extinction vortex)’ ….. How does introgression influence this ‘extinction vortex’?
Lines 56-62 – here the authors start off talking about the potential genetic effect of re-stocking programs on the wild populations of the taxon being stocked – which is, again, something that they are not examining in this study. Then they refer to the discussion in Allendorf et al. (2001) about a different issue – the effect of introgressive hybridization between introduced and native trout taxa (something that IS relevant to this study). Then they return to the genetic effects of stocking conspecifics, this time referring specifically to percichthyids.
Lines 63-80: as a reader, I was left with no clear understanding of why the Murray Cod and Trout Cod are considered distinct species. Are there morphological and genetic differences? Are there differences in habitat and distribution?
Lines 81-85: I would like to see some indication (on Fig 1 or elsewhere) where these fingerlings were released – this would make it easier to interpret the results and discussion. Did these stocked fish derive from the same hatchery that provided the Trout Cod reference larva? Where did the adults that produced these fingerlings originate?
Line 115: one larva, not one larvae.
Line 116: were these Murray Cod samples adults?
Line 121: as this appears to be relevant only to the Murray Cod samples (above), it would be clearer if this line were moved to the previous paragraph (i.e. line 117)
Line 127-129: all these references are unnecessary as the extraction protocol is subsequently described in detail. Alternatively – retain the reference and reduce the procedural detail given here.
Line 148-150: it would be helpful to point out here that DArTseq is a version of RADseq, as most readers will be more familiar with this term (it appears to be very similar to ddRAD with paired-end sequencing but without a size selection step (?), Peterson et al. 2012, PLoS ONE 7(5): e37135).
Line 150: ‘DArtseq represents a new implementation’ – this approach is not really new any more.
Line 168: was there a fragment size selection step? (presumably not?)
Line 178: I don’t quite understand what is going on with these “fastqcoll” files – was information on read number retained for SNP calling?
Lines 193-195: essentially, this repeats the information in lines 179-191.
Line 196: ‘..”1” denotes the presence and “0” denotes the absence of a marker in a genomic representation..’ Surely in this case 0, 1 represent the two alternate SNP alleles, not presence/absence?
Line 223: ‘Of the 21076 alleles…’ what does this mean? Are the authors trying to say the 21,076 variants were identified before quality control steps and 12,299 retained for further analysis?
Additionally: SNP genotypes for all individuals need to be made available (e.g. in a public repository) so that analyses can be replicated.
Line 230: better to give the web address for Tableau here, rather than an uninformative reference.
Line 244: ‘The length of the DNA fragments for each SNP in this sequencing was 69 base pairs.’ What does this mean? Are the authors talking about the average read length of the sequenced fragments?
Line 234: How many possible parental hybrid classes did the authors define for the NewHybrids analysis? (it looks like six).
Line 241: were the reference individuals considered to be part of the mixture, or not?
Line 247: presumably, the authors have decided the hybrid classes on the basis of the NewHybrids results, but these are not detailed here (where are they?).
Line 248: this is already stated in the methods.
Line 246: How many clusters did the K-means clustering find?
Line 247: technically, this is a DAPC plot, not a PCA plot.
Line 253: the mtDNA sequencing does not really ‘confirm hybridization’.
Line 255: ‘F1 backcross’?
Line 260: the authors should detail in the text (rather than as a footnote to Table2) how they identified this apparent ‘third generation cross’. Additionally, surely it is a MCBX x MC, not a MCBX x BX? Further, is it possible that this mitonuclear discordance is due to historic interbreeding between the previously sympatric species, rather than recent interbreeding involving stocked Trout Cod?
Table 2: it would be useful to include collection site/year on this table.
Discussion: in general, I think this could be shorter and more focussed.
Line 287: I don’t really understand this line. All successful mating events involving Trout Cod were heterospecific - in no way could these results be considered indicative of either ‘re-introduction’ or ‘recovery’ success. Conversely – they indicate that the two-decade re-introduction attempt for Trout Cod has failed miserably.
Lines 297-302: this is a list of all the possible outcomes of interspecific hybridization (again, it reads like a conservation genetics textbook). Instead, I would like to see a more focused discussion of what might be the potential outcomes of hybridization in THIS system, given that the propagule pressure from Trout Cod is relatively low and the two species previously co-existed in this system.
Line 303-312: this is not an either/or question; heterosis and outbreeding depression are expected to co-occur where there is ongoing introgressive hybridization between two populations.
Line 319 onwards: could the interspecific hybridization also be driven by changes in Trout Cod behaviour due to captive rearing or broodstock, or the stocked Trout Cod coming from a different population than the one that was originally present in the river?
Line 342 - 347: If the Trout Cod stocking program began above the Gigerline Gorge, why were larvae not sampled from this location?
Line 355: when was this remnant Trout Cod population extirpated? Again, is it possible that some of the hybrid larvae identified could have their ancestry from remnant wild Trout Cod rather than stocked individuals?
Line 390: ‘however’ does not seem like the correct word here.
Line 395: again, surely it’s most likely to be a MC backcross X MC?
Line 398: the problem is low statistical power to detect these later generation individuals, so larger samples would not fix this, but more species diagnostic SNPs/ more sensitive analytical techniques might.
Line 408: stocking greater numbers of Trout Cod may reduce introgression if the hybridization is due to difficulty in finding suitable mates – however it is just speculation whether or not this is the case. If the hybridization is due to some other factor (e.g. changes in Trout Cod behaviour due to captive rearing) then stocking in greater numbers might just increase the proportion of hybrid offspring.
Line 411: Comment on method – this section should be deleted, the utility of combining nuclear and mtDNA for investigating hybridization is already known, as is the utility of RAD genotyping-by-sequencing approaches (and this is not ‘phylogenetic work’).

---

## Round 0.2 · Minor Revisions

The authors are commended for their previous revisions, however, I had difficulties tracking down the changes made by the authors to their manuscript, given that they did not provide a response letter that clearly identified the new line numbers in relation to the modifications. There are a few outstanding issues that need to be dealt with in an additional revision:

1- Concerning one of your answers to reviewer 2: ‘Time constraints limit the number of replicates we can do. We have done 5, and there is extremely close agreement between each replicate. The single case where the replicates are not 100% concordant is highlighted in the results section and discussed.’

This is not a valid justification – please make at least 10 replicates and also include confidence intervals on the estimates – and please clarify the results related to the replicates as I could not conclude on the validity of the replicates based on what was currently provided.

2-Also - Concerning one of your answer to reviewer 3: ‘I guess so but we are less than intimately familiar with the genome reduction pipeline due to parts of it being proprietary.’

I really don’t understand what you mean here, and related to this point I still don’t understand the section on lines 217-220; please clarify.

3-Line 123: no need to include the page number.

3- Lines 311-316 are part of the discussion NOT the results.

4- you should thank the reviewers in Acknowledgements

---

## Round 0.3 · accepted · Accept

Thank you for performing these last required revisions.